# DC3DO: DIFFUSION CLASSIFIER FOR 3D OBJECTS

## ABSTRACT

Recent advancements in deep generative models, particularly diffusion models, have shown remarkable capabilities in generating high-fidelity 3D objects. In this work, we explore the application of diffusion models for 3D object classification by integrating the LION model with diffusion-based classifiers. Due to the availability of pretrained model weights, our study focuses on two categories from the ShapeNet dataset: chairs and cars. We propose DC3DO, a method that leverages the generative strengths of diffusion models for domain generalization in 3D classification tasks. Our approach demonstrates improved performance over a multi-view baseline, highlighting the potential of diffusion models in handling 3D data. We also examine the model's ability to generalize to data from different distributions, evaluating its performance on the IFCNet and ModelNet datasets. This study underscores the potential of using diffusion models for 3D object classification and sets the stage for future research involving more categories as resources become available.

## 1 INTRODUCTION

Recent advancements in deep generative models have yielded *state-of-the-art* (SOTA) performance in both classification and out-of-distribution (OOD) classification for images (Li et al., 2023a). These models are increasingly utilized for discriminative tasks, demonstrating superior effectiveness across various domains, including images (Huang et al., 2024), text (Han et al., 2022), and tabular data (Po et al., 2023; Huang et al., 2023). This progression builds upon the foundational work of Hinton (Hinton, 2007), inspired by Oliver Selfridge's "Pandemonium" model (Selfridge, 1958). While early research focused on generation within the image domain, it can be argued that understanding 3D structures and developing models capable of generating representations for 3D objects is intrinsic to comprehending data beyond point clouds. These representations can enhance downstream tasks in object classification and expand image classification by capturing the compositionality and alignment inherent in 3D object representations.

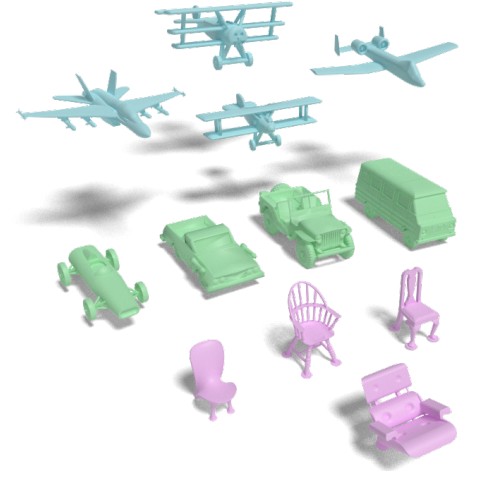

Figure 1: **Dataset classes for classification**. We performed 3D classification tests on cars, chairs, and airplanes. We used multi-view and point cloud representations only for chairs and cars.

The classification of 3D shapes is increasingly important in fields such as computer vision, robotics, and virtual reality. However, traditional methods often struggle to handle the complexity and variability inherent in 3D data. To address these challenges, we adopt a diffusion approach (Ho et al., 2020). Diffusion models (Sohl-Dickstein et al., 2015), a recent class of likelihood-based generative models, have shown significant promise in various tasks (Ramesh et al., 2022; Ho et al., 2022; Poole et al., 2022) by transforming random noise into coherent data samples through an iterative noising and denoising process. Recent advancements in diffusion models (Dhariwal & Nichol, 2021;

Preechakul et al., 2022; Shen et al., 2024) have demonstrated remarkable results in both generative tasks (Luo & Hu, 2021) and classification tasks (Meng et al., 2023; Lou et al., 2024). These models belong to a class of generative approaches that model the data distribution of datasets, similar to Variational Autoencoders (VAEs) (Kingma & Welling, 2014), Generative Adversarial Networks (GANs) (Wu et al., 2016; Chan et al., 2020), Energy-Based Models (EBMs) (Xie et al., 2021), and score-based models (Yang et al., 2019).

Therefore, a question arises: *Can we use diffusion models for 3D classification tasks?* More critically, given their remarkable ability to generate *original* objects beyond the initial dataset distribution (Zeng et al., 2022a; Nam et al., 2022; Tono et al., 2024; Luo & Hu, 2021; Zheng et al., 2023), how do these models perform on OOD data? While these models have excelled on standard benchmarks, they often struggle with novel OOD data—a limitation attributed to biased training datasets that fail to encompass the full spectrum of real-world possibilities (Jahanian et al., 2020). Despite these challenges, deep generative models can synthesize highly realistic and diverse images, objects, and text, and have demonstrated improved performance in zero-shot (Jain et al., 2021; 2022; Sanghi et al., 2023) and few-shot classification tasks (Shen et al., 2024).

In this work, we propose **DC3DO**, a novel approach that leverages diffusion models for the classification of 3D objects. By integrating Denoising Diffusion Probabilistic Models (DDPMs) (Ho et al., 2020), known for their immense representational capacity, into the 3D domain, we address the inherent complexity of 3D data. Traditional methods often fall short in effectively handling 3D data, highlighting the need for innovative approaches that can manage its unique characteristics. Three-dimensional data can be represented in various formats, including point clouds (Yu et al., 2021; Luo & Hu, 2021; Zhou et al., 2021), voxels (Choy et al., 2016; Wu et al., 2017), signed distance functions (Tono et al., 2024; Nam et al., 2022; Zeng et al., 2022b), and multi-view projections (Su et al., 2015). In our approach, we adopt the commonly used representation of point clouds, following works (Zhou et al., 2021; Zeng et al., 2022a), and integrate them with latent representations (Nam et al., 2022) within the framework of diffusion models. Due to the availability of pretrained model weights from LION, our study focuses on two categories from the ShapeNet dataset (Chang et al., 2015): *chairs* and *cars*. This integration allows us to utilize the strengths of these representations while leveraging the generative capabilities of diffusion models.

Our method, **DC3DO**, focuses on exploiting the generative power of diffusion models for zero-shot classification (Li et al., 2023b). To evaluate its effectiveness, we compare our approach against a direct extension of its 2D counterpart applied to images (Li et al., 2023a). For a fair comparison, we adapt MVCNN (Su et al., 2015), which traditionally employs a view pooling method, into a U-Net-based architecture compatible with diffusion classifiers. In today's dynamic data landscape, the ability to classify data into previously unseen categories—such as architectural structures (Tono et al., 2020; 2021; Stanislava et al., 2021)—is of paramount importance. Diffusion models, with their inherent generative strengths, are particularly well-suited for this challenge. By moving beyond traditional 2D prior models (Liu et al., 2023c;b;d) and incorporating the LION model (Zeng et al., 2022a), renowned for generating high-fidelity 3D shapes, we enhance the effectiveness of the diffusion classifier in performing discriminative tasks, particularly in 3D object classification.

Therefore, our contributions are as follows:

- **Novel method for 3D shape classification:** We present **DC3DO**, a diffusion model-based method for classifying 3D shapes, addressing the limitations of traditional classification methods when applied to 3D data.

- **Comparative analysis:** We compare our method against multi-view 3D representations using a 2D diffusion classifier Li et al. (2023a). We adapt MVCNN (Su et al., 2015), which utilizes a view pooling method, into a U-Net based architecture compatible with diffusion classifiers. This allows for a fair comparison between multi-view representations and our proposed method.

- **Evaluation of domain generalization:** Through empirical analysis, we demonstrate that our method maintains strong performance on OOD datasets, focusing on domain generalization within the chairs category. This showcases the adaptability and generalization capabilities of our approach beyond the training data.

## 2 RELATED WORK

Multimodal large language models (LLMs) strengths are leveraged in many current works (Qi et al., 2024; Ji et al., 2024; Xu et al., 2023; Guo et al., 2023). LLMs can handle diverse tasks through conversational interaction, specifically in the context of 3D objects. Typically, this is achieved by training a 3D shape encoder and aligning it with other modalities (e.g., text, images, and audio). The entire pipeline is then fine-tuned during an instruction-tuning phase, resulting in a model that is better aligned with user requests for specific 3D tasks. This fine-tuning stage is conducted using synthetic datasets or captioning datasets. These approaches highlight the vast potential of integrating 3D shapes into foundation models, although they still necessitate the fine-tuning of large models. Other methods, such as 3DAxiesPrompts (Liu et al., 2023a), enhance images and prompts with additional artifacts to be able to exploit the 2D vision abilities of LLM for 3D objects.

PEVA-Net (Lin et al., 2024) employs a pre-trained CLIP model in a multiview pipeline to classify 3D objects in zero-shot or few-shot environments. It leverages CLIP's zero-shot classification abilities for each view of the 3D object, subsequently aggregating these results to make the final prediction. Although this approach effectively exploits the zero-shot capabilities of vision-language models (VLMs), transforming 3D shapes into multiview images is an oversimplification that can lead to suboptimal results.

TAMM (Zhang et al., 2024) demonstrates that when aligning 3D object representations with other modalities, the image modality contributes less significantly than the text modality. To address this, their method learns to separate visual features from semantic features within the 3D object representation, enabling a more effective alignment with the other modalities and enhancing performance in downstream tasks. These findings suggest that the alignment between modalities for integrating 3D representations into existing methods can sometimes be inadequate (Xue et al., 2024). Regarding 3D representation learning, Zhang et al. (2022) takes a different approach and incorporates 2D guidance. Their work, dubbed I2P-MAE (Zhang et al., 2022), learns advanced 3D representations, achieving state-of-the-art performance on 3D tasks and significantly lowering the need for large-scale 3D datasets. On the contrary concurrent work, DiffCLIP (Shen et al., 2024) demonstrates that the integration of CLIP and diffusion models for 3D classification facilitates zero-shot classification, achieving state-of-the-art results. This methodology utilizes a pre-training pipeline that incorporates a Point Transformer for few-shot 3D point cloud classification, wherein the CLIP model extracts style-based features of the class, synergistically combined with image features. While DiffCLIP (Shen et al., 2024) used Point Transformer, we used LION(Zeng et al., 2022a), a latent point-voxel representation that leverages a hierarchical two stages diffusion process with state-of-the-art generative performances(Liu et al., 2019; Zhou et al., 2021). Following the line of latent and implicit representations, Yu et al. (2023) used a Classifier Score Distillation (CSD) method, which utilizes an implicit classification model for generation.

## 3 METHODOLOGY

We present and compare two distinct approaches for 3D object classification: Multi-View Diffusion Classifier (MVDC, Section 3.1) and DC3DO (Section 3.2). MVDC adapts the *Diffusion Classifier* (Li et al., 2023a) to handle multiple 3D views by implementing a majority vote mechanism across different perspectives. This serves as an alternative to the widely-used MVCNN, integrating diffusion models into the multi-view classification framework. DC3DO combines the generative model LION (Zeng et al., 2022a) with diffusion-based classification to enable domain generalization of complex 3D shapes, focusing on categories such as cars and chairs due to the availability of pre-trained models. By incorporating LION's high-fidelity 3D generation capabilities, DC3DO aims to enhance classification performance on shapes not seen during training.

Due to resource constraints and the availability of pretrained models, we focus our comparative analysis between MVDC and DC3DO, providing insights into the effectiveness of diffusion-based methods in 3D classification.

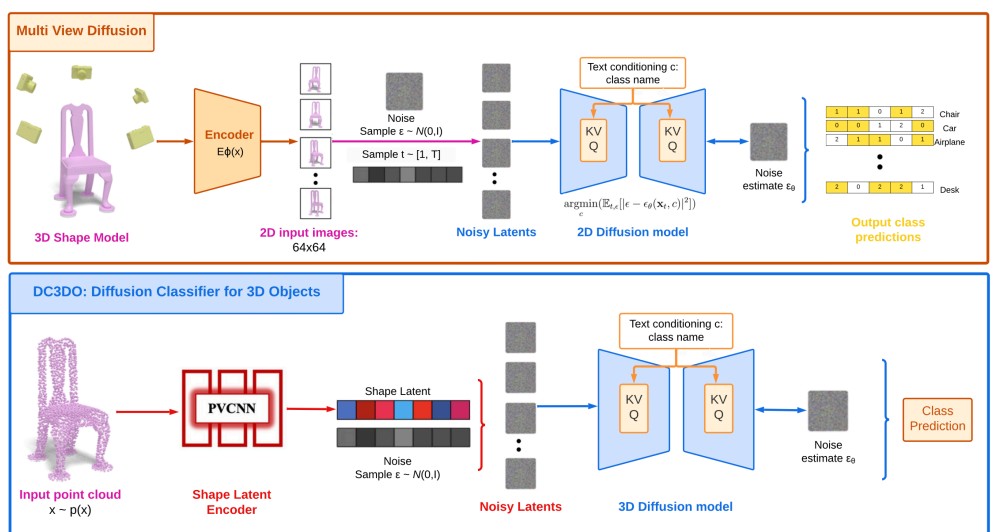

Figure 2: **Methods comparison**. We extended the *Diffusion Classifier* (Li et al., 2023a) paper to a multi-view (Su et al., 2015) settings and we compare with our DC3DO model, based on LION (Zeng et al., 2022a)

## 3.1 MULTI-VIEW DIFFUSION CLASSIFIER (MVDC)

3D objects can be effectively represented as a series of images, providing a straightforward baseline for extending previous work (Li et al., 2023a) to the 3D domain. By rendering multiple views of the same object, we can adapt existing diffusion-based classification techniques for 3D shapes.

We utilize the ShapeNet dataset (Chang et al., 2015), selecting a subset of $N$ models per class to balance computational demands and dataset diversity. Each 3D object is captured from $M$ distinct viewpoints, spaced at regular intervals around a horizontal circle encircling the object. This results in $M$ images per object, denoted as $X = \{X_1, X_2, \ldots, X_M\}$, where each $X_i$ represents a specific viewpoint.

Each view $X_i$ is independently processed by the diffusion-based classification model. The classification function $f(\cdot)$ assigns a prediction $y_i$ to each image:

$$y_i = f(X_i), \quad \text{for } i = 1, 2, \ldots, M \tag{1}$$

Unlike MVCNN (Su et al., 2015), which aggregates features from all views into a single global representation through view pooling, MVDC maintains separate predictions for each view. This approach allows us to utilize the pre-trained 2D diffusion classifier directly.

The final classification decision, denoted as $y^*$, is determined using a majority voting scheme:

$$y^* = \text{mode}(y_1, y_2, \ldots, y_M) \tag{2}$$

This majority vote approach preserves the individual predictions from each view, ensuring simplicity and interpretability while maintaining the architectural integrity of diffusion models.

**Feature Representation.** Each 2D image $X_i$ is processed and encoded into a feature map $\mathbf{i} \in \mathbb{R}^{H \times W \times C}$, where $H$, $W$, and $C$ represent the height, width, and number of channels of the image, respectively. (In our experiments, the highest image resolution used is $512 \times 512 \times 3$. Due to computational limitations, we also experiment with lower resolutions, which may affect classification performance.)

While MVCNN aggregates multi-view information into a unified global feature vector via view pooling, MVDC independently classifies each view and employs a majority vote to determine the

final class label. This distinction allows us to evaluate the effectiveness of diffusion-based classification methods in handling multi-view 3D data without feature aggregation, but may also limit the model's ability to integrate information across views, potentially impacting performance.

## 3.2 DIFFUSION CLASSIFIER FOR 3D OBJECTS (DC3DO)

DC3DO represents the primary contribution of our research, combining the LION model (Zeng et al., 2022a) with the diffusion classifier (Li et al., 2023a) for domain generalization of complex 3D shapes such as cars and chairs. By integrating LION's capability to generate diverse 3D shapes with the diffusion classifier, DC3DO aims to achieve accurate categorization of 3D objects within the classes it was trained on.

DC3DO integrates LION's hierarchical latent space with the diffusion classifier framework. LION employs a latent representation to capture both global and local features of 3D structures. The global latent space $\mathbf{z}_0 \in \mathbb{R}^{D_z}$ encodes the overall spatial structure of the 3D shape, while the local point-structured latent space $\mathbf{h}_0 \in \mathbb{R}^{(3+D_h) \times N}$ captures detailed, fine-grained features for each point in the point cloud.

**Hierarchical Latent Space.** LION's hierarchical latent space encodes 3D point clouds $\mathbf{x} \in \mathbb{R}^{3 \times N}$, where $N$ represents the number of points, into two distinct latent representations:

- **Global Latent Space**: $\mathbf{z}_0 \in \mathbb{R}^{D_z}$ captures the overall structure and large-scale features of the 3D object.
- **Local Point-Structured Latent Space**: $\mathbf{h}_0 \in \mathbb{R}^{(3+D_h) \times N}$ represents detailed features for each point, including $xyz$-coordinates and additional latent dimensions.

**Encoding.** The 3D point cloud data $\mathbf{x}$ is encoded into the global latent space using LION's PVCNN encoder. The global latents effectively capture the object's shape and high-level features necessary for classification. Additionally, the global latent space is smaller than the local point-structured latent space, simplifying the diffusion process, which involves multiple inference steps per sample. By focusing on the global latent space for the diffusion process, we reduce computational demands and capture essential structural information for classification.

**Diffusion Process.** Once the 3D point cloud data $\mathbf{x}$ is encoded into the hierarchical latent space, the latent representations undergo a diffusion process. This involves iteratively adding Gaussian noise to the latents over $T = 1000$ diffusion steps:

$$\mathbf{z}_t = \alpha_t \mathbf{z}_0 + \sigma_t \boldsymbol{\epsilon}, \quad \boldsymbol{\epsilon} \sim \mathcal{N}(\mathbf{0}, \mathbf{I}) \tag{3}$$

$$\mathbf{h}_t = \alpha_t \mathbf{h}_0 + \sigma_t \boldsymbol{\epsilon}, \quad \boldsymbol{\epsilon} \sim \mathcal{N}(\mathbf{0}, \mathbf{I}) \tag{4}$$

where $\alpha_t$ and $\sigma_t$ are coefficients that control the scaling of the original signal and the noise added at each timestep $t$, respectively.

**Denoising and Classification.** A neural network conditioned on class labels $\mathbf{c}$ performs the denoising of the perturbed latent data, reversing the diffusion process to retrieve the latent representations $\hat{\mathbf{z}}_0$ and $\hat{\mathbf{h}}_0$ that approximate the original data distribution. The network aims to minimize the reconstruction error by approximating the posterior distributions $q_\phi(\mathbf{z}_0 \mid \mathbf{x}, \mathbf{c})$ and $q_\phi(\mathbf{h}_0 \mid \mathbf{x}, \mathbf{z}_0, \mathbf{c})$. For classification, we compute the class-conditional likelihoods based on the denoising process of the global latent $\mathbf{z}_0$, evaluating which class label $\mathbf{c}_i$ best explains the observed data.

The classification decision is based on evaluating the class-conditional likelihoods $p_\theta(\mathbf{x}_0 \mid \mathbf{c}_i)$ for each class $\mathbf{c}_i$:

$$p_\theta(\mathbf{x}_0 \mid \mathbf{c}_i) = \int_{\mathbf{x}_{1:T}} p(\mathbf{x}_T) \prod_{t=1}^{T} p_\theta(\mathbf{x}_{t-1} \mid \mathbf{x}_t, \mathbf{c}_i) \, \mathrm{d}\mathbf{x}_{1:T} \tag{5}$$

The final classification assigns the object to the class with the highest likelihood:

$$\mathbf{c}^* = \arg\max_i \, p_\theta(\mathbf{x}_0 \mid \mathbf{c}_i) \tag{6}$$

By comparing the likelihoods across the known classes, the model assigns the object to the class whose denoising process best matches the encoded data.

**Class-Conditioned Diffusion.** In our implementation, we condition the diffusion process directly on class labels rather than textual descriptions. By using class labels such as "*chair*" and "*car*", we guide the model to classify 3D objects based on learned class-specific data distributions. By leveraging class-conditioned diffusion, we utilize the generative capabilities of the LION model to enhance classification performance within the evaluated classes. Incorporating textual descriptions into 3D diffusion models remains an area for future exploration.

## 4 EXPERIMENTAL RESULTS

### 4.1 MVDC - 2D RESULTS

In our baseline evaluation, we employed MVDC on the ShapeNet dataset (Chang et al., 2015), focusing on three classes: cars, chairs, and airplanes. We selected a subset of $N = 200$ models per class and utilized $M = 6$ frontal views per object to balance computational demands and maintain dataset diversity. The limited number of views and models was due to resource constraints, which we acknowledge as a limitation of our study.

The MVDC process involves encoding 3D shapes into latent representations using a pretrained classifier model. Note that in our implementation, we did not use a VAE but adapted the diffusion classifier from (Li et al., 2023a). Gaussian noise is then added to these representations, and a diffusion model is employed for denoising and classification. This methodology captures intricate details of 3D shapes and categorizes them by adaptively selecting the most promising samples based on predicted errors, thereby optimizing overall classification performance.

Table 1: Classification accuracy (%) of MVDC and DC3DO on ShapeNet for cars and chairs. We performed the comparison only on the first 200 models per class, each with 6 views and 100 sampling steps. Due to computational constraints, we limited the number of sampling steps to 100, which may impact the overall accuracy.

| Method | Accuracy | |
|---|---|---|
| | **Car** | **Chair** |
| MVDC (100 models) | 65.7% | 32.3% |
| MVDC (200 models) | 64.8% | 31.5% |
| **DC3DO-100m (ours)** | **100%** | **36%** |
| **DC3DO-200m (ours)** | **100%** | **49%** |

As shown in Table 1, DC3DO outperforms MVDC in classifying cars and chairs. However, we acknowledge that the absolute accuracies, especially for chairs, are relatively low compared to standard baselines. This may be due to the limited number of views, low image resolution, and the simplicity of the majority voting mechanism in MVDC.

MVCNN (Su et al., 2015) utilizes 36 fixed cameras with objects placed in a canonical pose. The cameras are positioned at uniform intervals, with a $10°$ rotation between each, defined by their position parameters $(X, Y, Z)$. To manage computational constraints, we downscaled the images to $64 \times 64$ and reduced the number of views per object from 36 to $M = 6$. This reduction in image resolution and number of views likely contributed to decreased classification performance in our experiments. The selected views correspond to camera angles of $10°$, $20°$, $30°$, $340°$, $350°$, and $360°$, aligning with the ShapeNet view settings.

As discussed in (Shen et al., 2024), frontal camera positions generally yield higher accuracy. Therefore, we focused on these $M = 6$ specific camera positions for our experiments.

To compute the accuracy of multi-view classification, we employed a majority vote mechanism across the $M = 6$ views of each 3D mesh. Let $y_i \in \{0, 1\}$ represent the binary prediction for each view $X_i$, where 1 corresponds to the prediction "car" and 0 corresponds to "not car". In this setup, if the number of votes for "car" (i.e., $\sum_{i=1}^{M} y_i$) is greater than or equal to $M/2$, the object is classified as "car." For each class $c$, we performed a binary classification, distinguishing between class $c$ and

all other classes. The accuracy $A_c$ for each class $c$ is calculated as:

$$A_c = \frac{\text{Number of correctly classified objects in class } c}{\text{Total number of objects in class } c}.$$ (7)

Finally, the mean per-class accuracy $\bar{A}$ is computed by averaging the binary classification accuracies across all classes:

$$\bar{A} = \frac{1}{C} \sum_{c=1}^{C} A_c$$ (8)

where $C$ is the total number of classes.

We acknowledge that using binary classification may not fully capture the complexities of multi-class 3D object classification. Future work will involve implementing multi-class classifiers.

### 4.2 DC3DO INFERENCE

DC3DO utilizes publicly available pretrained model weights from LION (Zeng et al., 2022a), which were trained on specific classes within the ShapeNet dataset, specifically "chairs" and "cars". By using these pretrained models, we avoid the need for additional training, thereby conserving computational resources. Due to computational constraints, we set the number of diffusion steps ($t$) for both the Multi-View Diffusion Classifier (MVDC) and DC3DO to $t = 200$ steps to reduce computational load, which may impact the overall performance.

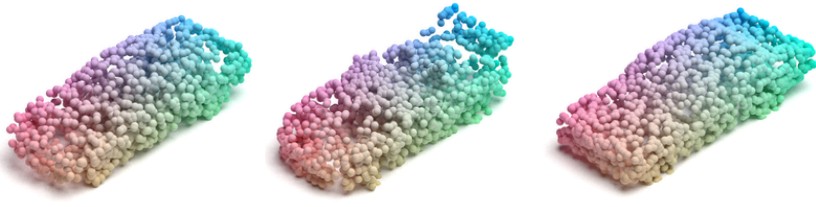

Figure 3: Examples of cars in ShapeNet with the highest classification confidence scores by DC3DO. The model classifies these shapes better since they have specific details that remove ambiguity form other cars in the dataset.

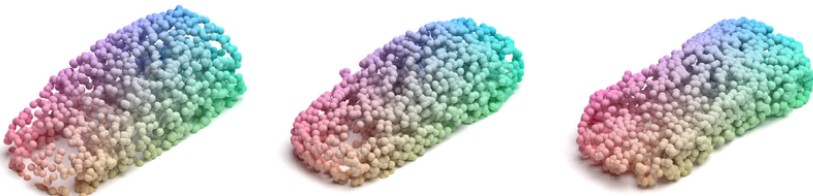

Figure 4: Examples of cars in ShapeNet with the lowest classification confidence scores by DC3DO. The overall shape of the car covers a large distribution space, since they all seem quite common car shape and topologies, probably also easy to confuse with other classes.

For our experiments, we employed a batch size of 1 for both the "cars" and "chairs" categories. Under the current computational settings, classifying each object takes approximately 20 seconds. We acknowledge that the small batch size and high computational time per sample limit the scalability of our approach. The classification process involves encoding the 3D point cloud data, applying the diffusion process, and performing denoising to predict the class labels.

As shown in Table 1, our DC3DO model significantly outperforms the 2D MVDC used as a baseline, demonstrating its superior effectiveness in classifying 3D objects. Additionally, Figures 3 and 4 show that DC3DO performs well on cars with distinctive features but struggles with cars

that have common shapes, leading to lower classification accuracies. Similarly, Table 2 illustrates that DC3DO achieves higher accuracy on chairs with unique designs but has difficulty accurately classifying chairs with more generic structures.

Table 2: Rendered images and classification accuracies of the top-performing and lowest-performing chairs in ShapeNet as classified by DC3DO.

| Best Chairs | | Worst Chairs | |
|---|---|---|---|
| Render | Accuracy | Render | Accuracy |
|  | 65.2% |  | 23.2% |
|  | 61.8% |  | 22.1% |
|  | 58.3% |  | 20.7% |

### 4.3 OUT-OF-DISTRIBUTION CLASSIFIER PERFORMANCE

In this section, we evaluate the domain generalization capabilities of our integrated LION and diffusion classifier model on both in-distribution (ID) and out-of-distribution (OOD) data. The ID data is sourced from the ShapeNet dataset (Chang et al., 2015), which was used to train the LION model, while the OOD data comprises data samples from ModelNet (Wu et al., 2015), and IFCNet (Emunds et al., 2021) datasets that were not encountered during training.

We define two distinct test sets for evaluation:

**ID Test Set:** $\mathcal{D}_{\text{ID}} = \{(x_i, y_i)\}_{i=1}^{N_{\text{ID}}}$: Each $x_i$ is a 3D chair model sourced from the ShapeNet dataset, and $y_i$ is its corresponding class label within the ShapeNet categories. This test set assesses the model's performance on data drawn from the same distribution as the training set.

**OOD Test Set:** $\mathcal{D}_{\text{OOD}} = \{(x_j, y_j)\}_{j=1}^{N_{\text{OOD}}}$: Each $x_j$ is a 3D chair model sourced from ModelNet or IFCNet datasets, and $y_j$ is its corresponding class label. Although both $\mathcal{D}_{\text{ID}}$ and $\mathcal{D}_{\text{OOD}}$ contain chairs, the OOD data originates from different sources, introducing a distributional shift despite sharing the same category. This test set evaluates the model's ability to generalize to data from unseen distributions.

Table 3: **Classification accuracy (%) and OOD generalization results for DC3DO on ShapeNet, ModelNet, and IFCNet.** We performed the evaluation for two configurations: DC3DO-100m and DC3DO-200m. For the DC3DO-200m configuration, we report the average accuracies based on 200 subsample models, each processed with 200 sampling steps.

| Method | ID Chairs Accuracy | OOD Chairs Accuracy | |
|---|---|---|---|
| | **Shapenet** | **ModelNet** | **IFCNet** |
| **DC3DO-100m** | 36.72% | 32.81% | 27.92% |
| **DC3DO-200m** | 49.50% | 42.56% | 32.63% |

Our classification model, denoted as $f_\theta$, maps input 3D chair models to their predicted class labels, i.e. $f_\theta : x \rightarrow \hat{y}$ where $x$ represents a 3D chair model, $\hat{y}$ represents the predicted class label, $\theta$ denotes the model parameters learned during training.

We measure the model's performance using the classification accuracy metric, defined as:

$$\text{Accuracy} = \frac{1}{|\mathcal{D}|} \sum_{(x,y) \in \mathcal{D}} \mathbb{I}\big(f_\theta(x) = y\big), \qquad (9)$$

where $|\mathcal{D}|$ is the total number of samples in the test set $\mathcal{D}$, $\mathbb{I}(\cdot)$ is the indicator function that returns 1 if the condition inside is true and 0 otherwise. $\mathcal{D}$ can be either $\mathcal{D}_{\text{ID}}$ or $\mathcal{D}_{\text{OOD}}$, depending on the evaluation context.

Table 3 presents the classification accuracy and OOD generalization results for DC3DO on ShapeNet (ID), ModelNet, and IFCNet (OOD). We evaluated two configurations: DC3DO-100m and DC3DO-200m. For the DC3DO-200m configuration, we report the average accuracies based on 200 subsample models, each processed with 200 diffusion steps. The DC3DO-200m configuration exhibits an improvement over DC3DO-100m in both ID and OOD settings. The increase in the number of diffusion steps and subsample models enhances the model's ability to generalize and accurately classify chairs from unseen distributions such as ModelNet and IFCNet. Due to computational constraints and the availability of pretrained model weights, we were unable to compare our method directly with standard baselines on OOD data. We acknowledge that such comparisons would provide a more comprehensive evaluation of our approach, and we plan to address this in future work. Our findings suggest that while DC3DO shows potential in domain generalization, further optimization and evaluation on more categories are needed to assess its effectiveness fully.

### 4.4 ABLATION STUDIES

To gain deeper insights into the contributions of different components in our model, we conducted ablation studies for image resolution and quantifying its impact on performance.

We conducted experiments with various image sizes to study their impact on inference time and classification performance. First, we confirmed that the inference time grows exponentially with larger image sizes. For a $512 \times 512$ resolution image and 500 sampling steps, the processing time was approximately 1.5 minutes per image, making it infeasible to evaluate at larger scales. Moreover, when we reduced the image size to $S = 64 \times 64$ or $S = 128 \times 128$, the classifier's performance degraded significantly. The model exhibited a tendency to collapse, consistently predicting a single class $c$ regardless of the input views, suggesting that the classifier lost its ability to differentiate between classes under reduced image resolutions.

Table 4: **Ablation studies about image resolutions**. Inference time and accuracy analysis of MVDC model on three classes from Shapenet: airplane, car, and chair. The sample size is fixed at 200 steps.

| Image Resolution | Inference Time | Accuracy | | |
|---|---|---|---|---|
| | | Airplane | Car | Chair |
| $64 \times 64$ | 1h03m | 66.7% | 64.8% | 31.5% |
| $128 \times 128$ | 2h13m | 33.7% | 66.7% | 67.0% |
| $256 \times 256$ | 7h05m | 99.3% | 98.7% | 99.% |

## 5 LIMITATIONS

One of the primary limitations of our approach is the computational cost. The 3D diffusion process currently requires approximately 20 minutes per object on a T4 GPU, making it a time-intensive task. Similarly, the multi-view approach, while effective, is also relatively slow due to the independent processing of each view. Additionally, our experiments were limited to two categories—chairs and cars—due to the availability of pretrained models.

Regarding the MVDC, a significant limitation is that the views are processed individually and then aggregated through a majority vote, rather than being combined into a global latent vector as in the approach used by MVCNN (Su et al., 2015). This method of independent view processing may not fully capture the holistic structure of 3D shapes, which could be better represented through a more integrated multi-view approach. redMoreover, the majority vote mechanism is simplistic and may not effectively leverage the combined information from multiple views, potentially limiting classification accuracy.

Due to time and computational constraints, we limited our experiments to 200 shapes per category. Future work could expand to more categories and objects, and enhance the diffusion and aggregation methods to improve scalability and performance. Access to more powerful GPUs and resources would enable more comprehensive experiments.

## 6 Discussion and Future Work

The classification accuracy on ID data indicates that the model effectively captures the distinguishing features of various 3D objects within the evaluated categories. However, the performance on OOD data reveals areas for improvement, highlighting the challenges of generalizing to unseen distributions.

The hierarchical latent space of LION played a crucial role in accurately representing both global and local features of 3D shapes, contributing to the model's overall performance compared the multi-view (see Section 5 for more details). The diffusion process further enhanced the model's ability to denoise and classify complex 3D structures, providing a reliable mechanism for domain generalization.

Our results highlight the potential of integrating generative models like LION with diffusion classifiers for advanced 3D shape analysis and classification tasks, particularly in scenarios involving diverse and unseen data. However, the limitations identified, such as computational costs and challenges in capturing holistic structures, suggest that further optimization is necessary to fully realize this potential.

In fact, in this work, we delved into 3D diffusion models and present our method that enables domain generalization of 3D shapes in a robust manner.

For future work, we aim to explore the integration of 3D diffusion capabilities with state-of-the-art multimodal methods such as ULIP-2 (Xue et al., 2024), combined with PointBERT (Yu et al., 2021) architectures similar to the concurrent work (Shen et al., 2024). We believe this will enhance the performance of these architectures and make them more capable of comprehensive 3D understanding. Additionally, future research will focus on improving the efficiency of the diffusion process, expanding the range of evaluated categories, and implementing more sophisticated aggregation mechanisms to better capture the holistic structure of 3D objects.

## 7 Conclusion

In this paper, we proposed **DC3DO** in which we seamlessly integrate LION Zeng et al. (2022a) with a diffusion classifier Li et al. (2023a) to achieve accurate classification of 3D cars and chairs. The model's success is driven by the hierarchical latent space and diffusion process, when combined enable precise representation and classification of complex 3D shapes from the ShapeNet dataset Chang et al. (2015). Our method demonstrates a 12.5% improvement on average compared to multi-view methods, highlighting the potential of generative models in 3D object classification. In future research, we would like to adapt generative models to discriminative tasks for enhanced classification and regression performance as well as incorporate group structure into the diffusion model for improving data efficiency.

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
