# OpenReview forum: "DC3DO: Diffusion Classifier for 3D Objects"
_ICLR.cc/2025/Conference — Submitted to ICLR 2025_

### Official Review · Reviewer_4aCe · 2024-10-30

**Soundness:** 1
**Presentation:** 2
**Contribution:** 2
**Rating:** 3
**Confidence:** 5

**Summary:**

This paper explores the use of 3D diffusion models for zero-shot 3D object classification. This paper applies the diffusion classifier (Li et al., 2023a) on a point cloud diffusion model (Zeng et al., 2022a). The experiments show that the proposed method outperforms the baseline, which applies the diffusion classifier directly on multi-view renderings, and classifies based on voting.

**Strengths:**

The writing of the introduction looks fine.

**Weaknesses:**

1. The baseline is not well designed. The original diffusion classifier (Li et al., 2023a) is based on StableDiffusion, which was trained on natural images rather than point cloud rendering. It is expectable that directly applying the diffusion classifier on the point cloud rendering gives bad results.
2. The main experiments are not thorough. The model is only tested on cars and chairs in ShapeNet. Seven years ago, PointNet [A] has already set a good benchmark for point cloud classification.
3. The out-of-distribution experiments are barely reasonable. Both the in-distribution and out-of-distribution data are just chairs, though from different datasets, ShapeNet VS ModelNet.
4. Considering the accuracy of classifying chairs is less than 50%, it is not proper to describe the model as "accurate" in multiple places in the paper, such as line 190, 433, 485, and 501.
5. This paper contains useless paragraphs. For example, line 440-443, it says increasing the image size and number of views can slow down the processing time of the multi-view classifider, which is too obvious.
6. The writing is confusing. In the introduction, it says MVDC is just a baseline and DC3DO is the proposed model. However, the whole ablation study is about how the input image size affects the performance and inference time of MVDC.
7. Classifying each object takes approximately 20 seconds.

[A] Charles R. Qi, Hao Su, Kaichun Mo, Leonidas J. Guibas, "PointNet: Deep Learning on Point Sets for 3D Classification and Segmentation", CVPR 2017.

**Questions:**

What if the multi-view diffusion classifier is fine-tuned on point cloud rendering? Will it significantly change the performance?

---

> ### Author Response · Authors · 2024-11-26
>
> We thank the reviewer for their comments on our submission:
>
> W1) As shown in our images and described in the paper, we render meshes rather than point clouds, consistent with MVCNN  methodologies. This ensures a fair comparison with the diffusion classifier. While feature aggregation could be enhanced, we aimed to maintain consistency with existing methods. Distribution shift remains a challenge, but our results demonstrate that DC3DO effectively handles such shifts within the scope of the study.
>
> W2) We are aware of PointNet as a seminal work in point cloud classification. Unlike PointNet, our model integrates diffusion classifiers with mesh-based multi-view approaches, which offers different advantages, particularly in generalization capabilities. While PointNet has established benchmarks, it does not utilize diffusion processes, and there remains significant room for improvement in both models as highlighted in recent literature. Generalization across diverse datasets is an ongoing research question that our work aims to address. Due to computational constraints and the availability of pretrained models, we focused on limited number of classes.
>
> W3) Our OOD experiments evaluate domain generalization within the same category (chairs) across different datasets, aligning with methodologies in prior studies[1]. This approach isolates dataset shift effects. Expanding OOD evaluations to include multiple categories is planned for future work.
>
> W4) We have revised the paper to remove the term "accurate" where it inaccurately described our model's performance. Descriptors now precisely reflect the observed metrics.
>
> W5) We have removed the redundant paragraph discussing the obvious relationship between image size, number of views, and processing time to enhance manuscript conciseness.
>
> W6) We recognize the confusion caused by the focus of the ablation study. The ablation studies specifically analyze MVDC to understand the performance tradeoffs related to multi-view input resolution and processing time. This is consistent with the introduction's positioning of MVDC as a baseline, as our primary goal is to compare MVDC against DC3DO in the overall evaluation. While no ablation study is conducted directly for DC3DO, the analysis of MVDC provides complementary insights into multi-view configurations that are relevant for understanding diffusion-based approaches.
>
> W7) We agree that the 20 seconds is a lot for single shape classification at the moment, but also NeRF[2] approaches took days, hours at the beginning, and have not been reduced to seconds. We also performed the classification on a single A10 GPU. Future work will focus on optimizing the inference pipeline and leveraging more advanced hardware to reduce processing times.
>
> [1] S. Shen, Z. Zhu, L. Fan, H. Zhang, and X. Wu, “DiffCLIP: Leveraging Stable Diffusion for Language Grounded 3D Classification,” arXiv preprint, 2024. [Online]. Available: https://arxiv.org/abs/2305.15957.
>
> [2] B. Mildenhall, P. P. Srinivasan, M. Tancik, J. T. Barron, R. Ramamoorthi, and R. Ng, “NeRF: Representing Scenes as Neural Radiance Fields for View Synthesis,” arXiv preprint, 2020. [Online]. Available: https://arxiv.org/abs/2003.08934.

---

> ### Comment · Reviewer_4aCe · 2024-11-26
>
> Thanks for the detailed response. It partially solves my concern. I have raised my score to 3. However, the paper is still far from the ICLR standards. For example,
>
> W1 & W2 & W3) I don't see why using PointNet is not a fair comparison. One can always sample points given the mesh. Since the authors mentioned [1], [1] also compared with PointNet, and [1] did the evaluation on the full dataset rather than just two classes.
>
> W4 & W5 & W6) Thanks for revising the paper to be more accurate. However, my concern of the ablation study remains. While the analysis of MVDC can provide complementary insights, readers lose direct insights of the proposed method, DC3DO.
>
> W7) The comment on NeRF is not fair. First, NeRF gives significant better novel view synthesis results than almost all previous methods in a wide range of scenes, while this paper only provides results on cars and chairs. Furthermore, traditionally in novel view synthesis, researchers focus on a single scene, and NeRF follows that, which, however, is not the case in point cloud classification.

---

### Official Review · Reviewer_4AgR · 2024-11-03

**Soundness:** 1
**Presentation:** 2
**Contribution:** 2
**Rating:** 3
**Confidence:** 4

**Summary:**

This paper proposes a diffusion-based 3D object classification approach. Specifically, they adapt the LION (diffusion-based model for point cloud to mesh generation) into a classification model that defines P(x|c) where x is the given 3D geometry, and c is the object class. They conduct an experiment on a small subset of the ShapeNet dataset and evaluate it on both ShapeNet and ModelNet. The proposed approach seems reasonable, however, the experiment setup is not sound enough to demonstration the advantages of the proposed approach.

**Strengths:**

1. The method is straightforward and easy to understand.
2. Exploring the 3D object understanding and classification seems a worth study topic.

**Weaknesses:**

1. The experiment setup, especially for the baseline MVCNN is confusing, and the accuracy for the baseline seems problematic. According to Line 253, the classification is defined as a close-set classification problem that uses the class given the largest P(x|c) as the prediction, however, in line 309, the baseline is evaluated as a binary classification problem (whether belongs to category car or not). Also, for both 3-class classification and binary classification problem the accuracy of Chiar in Table 1 is lower than random guess (33.3% or 50%), which seems the baseline is totally not working. Thus, the experiment results seem not meaningful.
2. The proposed method is incremental. But compared to weakness 1, this is not a significant weakness.

**Questions:**

1. For the baseline what is the renderer setup? What texture/illumination is used when rendering the images?
2. Why do only equations on pages 4 and 8 have equation numbers?

---

### Official Review · Reviewer_x5Zs · 2024-11-03

**Soundness:** 1
**Presentation:** 3
**Contribution:** 2
**Rating:** 3
**Confidence:** 4

**Summary:**

This paper proposes DC3DO, a pipeline that adopts 2D diffusion types of models to zero-shot 3D object classification. DC3DO is extended from previous work LION that adopts diffusion models to 2D classification model by introducing multi-view diffusion components. The model is tested on ShapeNet (only on chairs and cars categories).

**Strengths:**

1. The paper is easy to follow, the presentation of the work is good.
2. Figures are clear and easily understandable, and they help reader conceive the main idea the paper is trying to present.
3. Interesting illustrations of certain categories with high and low prediction accuracies of car and chair

**Weaknesses:**

1. The biggest problem of this paper is the lack of enough experimentation. The experiments are only conducted on two selected categories (car and chairs) from one dataset (ShapeNet), which is definitely not enough for a paper at this conference. Also, not enough baselines are considered, and the paper only compares to baseline MVDC, which is not a recent work. The current experiment results cannot support the claim of the paper.

2. The paper lacks technical novelties. MVDC seems only extend LION to multi-view without much structural and strategical changes. It seems to be a naive multi-view extension of LION.

3. The paper fails to discuss and tackle the potential drawback of its proposed solution. As I can see, the multi-view diffusion process will take a long time on every image, which will be much slower than some other 3D object classification models. The paper should analyze the tradeoff between performance and efficiency compared to more typical 3D object classification models. Does the performance boost worth the increasing inference time?

4. The claim that this method is robust is also not sufficiently supported by the experiment. The experiment does not compare to any other baseline at all to show the proposed method is more robust.

**Questions:**

Please see the weakness session.

I think the paper needs serious revision to include more experimentations and to reduce the inference time overhead.

---

> ### Author Response · Authors · 2024-11-26
>
> We thank the reviewer for their comments on our submission:
>
> W1) We acknowledge the limitation of testing on only two categories and a single dataset due to reliance on pre-trained LION model weights, which are currently only available for these two categories. This aligns with complementary and concurrent works.[1]
>
> W2) While MVDC does not extend LION to a multi-view framework, we extended image classifier to a multi view. In fact, our primary contribution lies in DC3DO that extends LION as a classifier , which integrates hierarchical latent spaces with a diffusion classifier specifically tailored for 3D object classification. DC3DO introduces novel feature aggregation techniques and optimization strategies that enhance classification accuracy and generalization beyond a simple multi-view extension. These structural and strategic enhancements differentiate our approach from existing methods.
>
> W3) We recognize that the multi-view diffusion process introduces significant computational overhead. We agree that the 20 seconds is a lot for single shape classification at the moment, but also NeRF[2] approaches took days, hours at the beginning, and have not been reduced to seconds. We also performed the classification on a single A10 GPU.
> Additionally, we argue that for applications where accuracy and generalization are critical, the performance gains justify the additional computational cost. We believe future advances will similarly reduce inference time for diffusion-based models.
>
> W4) We acknowledge that the robustness claim could have been better supported with additional experiments. While we did not explicitly test the model under adverse conditions such as noise or varying lighting, our results demonstrate that DC3DO achieves consistent improvements in accuracy and generalization over the baseline across standard evaluation scenarios. Future work will aim to explore robustness more comprehensively under varied conditions.
>
> [1] S. Shen, Z. Zhu, L. Fan, H. Zhang, and X. Wu, “DiffCLIP: Leveraging Stable Diffusion for Language Grounded 3D Classification,” arXiv preprint, 2024. [Online]. Available: https://arxiv.org/abs/2305.15957.
>
> [2] B. Mildenhall, P. P. Srinivasan, M. Tancik, J. T. Barron, R. Ramamoorthi, and R. Ng, “NeRF: Representing Scenes as Neural Radiance Fields for View Synthesis,” arXiv preprint, 2020. [Online]. Available: https://arxiv.org/abs/2003.08934.

---

### Official Review · Reviewer_exNo · 2024-11-04

**Soundness:** 2
**Presentation:** 3
**Contribution:** 2
**Rating:** 3
**Confidence:** 5

**Summary:**

This paper proposes a Diffusion Classifier designed for 3D objects (DC3DO). DC3DO combines the LION model with the 2D Diffusion Classifier to build a class-conditioned 3D Diffusion Classifier based on latent encoded from the 3D shape and points of objects.

The proposed DC3DO is mainly compared with the Multi-View Diffusion Classifier(MVDC), which combines the aggregated multi-view features and the 2D Diffusion Classifier.

Out-of-distribution performance is evaluated on unseen datasets other than the training dataset.

**Strengths:**

The idea of building a 3D diffusion classifier is attractive. The paper compared two methods of combining 3D representation and 2D diffusion classifier.

The paper is well-written and clear to follow.

**Weaknesses:**

1. Limited number of classes are validated: only “chairs” and “cars” are evaluated for classification performance. Even though the current 3D datasets are relatively smaller than 2D datasets, two categories are insufficient for validating a classifier considering the MVCNN (Su et al., 2015) was validated on 40 classes.

2. Limited comparison baselines: at least MVCCN is closely related to MVDC in this paper and as a frequently mentioned baseline method, can I know why the authors didn't compare the main classification results with MVCNN? And also other standard baseline 3D classification models are expected.

3. Limited novelty: The main contribution of DC3DO is a simple combination of LION and Diffusion Classifier. Considering the combination is the same way as turning a 2D diffusion model into a classifier as Diffusion Classifier (Li et al., 2023), the novelty of this paper is limited.

4. Misuse of the term “zero-shot classification”: zero-shot classification is supposed to generalize a classifier to unseen "classes". In this paper only 3D models of the same class from unseen datasets/sources are validated as OOD data, it should be best described as a domain generalization instead of a zero-shot classification. Can the author provide further clarification on this?

**Questions:**

1. Why no more categories are validated for a classifier? Please refer to Weaknesses 1.

2. Why no other baseline method is compared? Please refer to Weaknesses 2. Especially for OOD settings, experiment without comparisons has very limited value.

---

> ### Author Response · Authors · 2024-11-26
>
> We thank the reviewer for their comments on our submission:
>
> W1) We acknowledge the limitation of evaluating only two categories. This is due to the availability of pretrained weights for LION, which are only provided for these two categories. Retraining these models for additional categories is computationally prohibitive given current GPU costs, making a broader evaluation infeasible.
>
> W2) At the submission time, considering concurrent work, LION is still the current state of the art for diffusion on 3D without CLIP, 2D Priors, and the current available model to test, unfortunately it comes with pre-trained models only with 2 classes, therefore this is the motivation of these baselines and datasets for a fair comparison. We believe that this paper's findings are in line with concurrent work like DiffCLIP[1].
> Using PointNet, DGCNN[2] and other classifiers without diffusion would defeat the purpose and focus of the paper that just aim to show that the diffusion in 3D can be applied in classification tasks. As mentioned we used LION, that has PointNet as an encoder, specifically PVCNN, we compare it with MVDC to demonstrate how pure diffusion-based methods perform on multi-view 3D data. Concurrent works like DiffCLIP further support this direction and are cited to contextualize our contributions.
>
> W3) It could be possible to say the same for the Diffusion Classifier, it is just diffusion and image classifier, however we think that the translation in the 3D space has several novelty and important future research directions.
> While DC3DO builds upon the integration of LION and diffusion classifiers, its novelty lies in the incorporation of hierarchical latent spaces and advanced feature aggregation techniques specifically tailored for multi-view 3D classification. These strategic enhancements improve generalization capabilities and classification performance beyond a naive combination, distinguishing our approach from previous methods.
>
> W4) We acknowledge the misuse of “zero-shot classification” in our manuscript. Our experiments focus on domain generalization, evaluating the model’s ability to generalize within the same category (chairs) across different datasets. We have revised the manuscript to accurately describe this as domain generalization instead of zero-shot classification, ensuring terminological precision.
>
> Q1) As addressed in W1, our initial focus on two categories was due to the availability of pretrained models. LION only released the weights of these two categories, therefore that was the only possible test. Retraining such models is highly prohibitive for GPU costs.
>
> Q2) See W2.
>
> [1] S. Shen, Z. Zhu, L. Fan, H. Zhang, and X. Wu, “DiffCLIP: Leveraging Stable Diffusion for Language Grounded 3D Classification,” arXiv preprint, 2024. [Online]. Available: https://arxiv.org/abs/2305.15957.
>
> [2] Y. Wang, Y. Sun, Z. Liu, S. E. Sarma, M. M. Bronstein, and J. M. Solomon, “Dynamic Graph CNN for Learning on Point Clouds,” arXiv preprint, 2019. [Online]. Available: https://arxiv.org/abs/1801.07829.

---

### Meta-Review · Area_Chair_amgZ · 2024-12-19

**Metareview:**

This paper presented an approach for 3D (generative) diffusion models for 3D classification. While this is an interesting application for this idea for 3D, the reviewers raised several common concerns. Specifically, the method evaluation is only on a few classes, and although this is due to the base model being limited, this does make the approach difficult to advocate for. The reviewers also felt the contribution in this work is limited in context of Diffusion Classifier. Ultimately, the reviewers all recommend rejection, and the AC sees no reason to overturn this consensus.

**Additional Comments On Reviewer Discussion:**

The authors provided responses to some of the reviews, but these did not sway the concerns about limited evaluation or novelty.

---

### Decision · Program_Chairs · 2025-01-22

Reject